# Molecular Mechanisms Underlying Qi-Invigorating Effects in Traditional Medicine: Network Pharmacology-Based Study on the Unique Functions of Qi-Invigorating Herb Group

**DOI:** 10.3390/plants11192470

**Published:** 2022-09-21

**Authors:** Minh Nhat Tran, Soyoung Kim, Quynh Hoang Ngan Nguyen, Sanghun Lee

**Affiliations:** 1Korean Medicine Data Division, Korea Institute of Oriental Medicine, Daejeon 34054, Korea; 2Korean Convergence Medical Science, University of Science and Technology, Daejeon 34113, Korea; 3Faculty of Traditional Medicine, Hue University of Medicine and Pharmacy, Hue University, Hue 49120, Vietnam; 4Center for Artificial Intelligence, Korea Institute of Science and Technology, Seoul 02792, Korea; 5AI Robotics, University of Science and Technology, Daejeon 34113, Korea

**Keywords:** Qi-invigorating herb, network pharmacology, unique compound, molecular mechanism

## Abstract

Qi-invigorating herbs (QIHs) are a group of herbs that invigorate Qi, the most vital force for maintaining the physiological functions of the human body in traditional medicine. However, the mechanism underlying the Qi-invigorating effects remains unclear. This study aimed to elucidate the unique mechanisms of QIHs based on unique compounds, using a network pharmacology approach. QIHs and their compounds were identified using existing literature and the TCMSP database, respectively. Subsequently, a method was proposed to screen for unique compounds that are common in QIHs but rare in other traditional herbs. Unique compounds’ targets were predicted using the TCMSP, BATMAN-TCM, and SwissTargetPrediction databases. Finally, enriched GO and KEGG pathways were obtained using DAVID to uncover the biomolecular functions and mechanisms. Thirteen unique compounds, mainly including amino acids and vitamins that participate in energy metabolism and improve Qi deficiency syndrome, were identified among the eight QIHs. GO and KEGG pathway analyses revealed that these compounds commonly participate in neuroactive ligand–receptor interaction and the metabolism of amino acids, and are related to the components of mitochondria and neuronal cells. Our results appropriately reflect the characteristics of traditional Qi-invigorating effects; therefore, this study facilitates the scientific interpretation of Qi functions and provides evidence regarding the treatment effectiveness of QIHs.

## 1. Introduction

The concept of Qi is commonly accepted as vital energy in constructing the human body and maintaining living activities in East Asian traditional medicine (TM) [1]. A lack of Qi leads to Qi deficiency syndrome, which is characterized by symptoms such as fatigue, shortness of breath, spontaneous sweating, pale tongue, and weak pulse [2], which are associated with chronic disease and sub-health pathogenesis [3], and significantly affect the quality of life [4]. Qi-invigorating herbs (QIHs) are a group of herbs that invigorate healthy Qi and that are used to improve the symptoms of Qi deficiency syndrome [2]. Recent studies have attempted to demonstrate the effects of various QIHs on mitochondria, the cellular “powerhouses,” to elucidate their energy-boosting effect [5,6]. Other studies have demonstrated the role of QIHs in fighting fatigue [7] and as antioxidants and immunostimulants [8]. However, owing to the multi-compound properties of QIHs and the immateriality of Qi, the biological mechanism of this herbal group has not yet been elucidated.

Recently, as an emerging discipline in medicine, medical bioinformatics has combined mathematics, information science, and biology in health research and has been used to investigate the interactional effects of both western medicine and TM [9]. Network pharmacology has been helpful in the study of functional networks of traditional herbs, providing insights into molecular mechanisms and improving drug design, which can further validate the value of TM in modern medicine [9,10]. Network-pharmacology-based studies showed that compounds of QIHs had high lipophilicity and unsaturated carbons [11], and QIHs promoted immune modules such as macrophage and lymphocyte proliferation [12]. These studies have provided a scientific foundation for research on QIHs; however, the unique mechanisms pertaining to specific functions of QIHs remain unclear.

In this study, we defined the unique function of the QIH group through the effect and metabolic pathways of its unique compounds based on the network pharmacology method. First, QIHs were identified using previously published literature, and their compounds were searched for in the database. Next, we proposed a method to screen unique compounds that represent specific features of the respective QIHs. Finally, the pathways of the unique compounds were predicted based on their targets using different bioinformatics databases, and a visualized network of a “herb-unique compound-target-pathway” was constructed to uncover the unique mechanisms underlying the Qi-stimulating effect of QIHs. This process can not only be used to scientifically interpret the mechanism of Qi-invigorating effects but also to provide evidence regarding the treatment effectiveness of QIHs. The study flowchart is shown in Figure 1.

## 2. Results

### 2.1. Identification of QIHs and Their Compounds

Using three representative textbooks written in different languages, 9 common QIHs were chosen among the 16 QIHs presented in “Zhong yao xue”, the 9 in “Boncho-hak”, and the 12 in “Chinese medical herbology and pharmacology” (Appendix A).

Chemical compounds of eight of the nine common QIHs were identified using the TCMSP database owing to the unavailability of Mel (feng mi). A total of 860 compounds were obtained (Appendix A), including 190 from Radix Ginseng (RG1, Renshen), 134 from Radix Codonopsis (RC, Dangshen), 87 from Radix Astragali (RA, Huangqi), 55 from Rhizoma Atractylodis Macrocephalae (RAM, Baizhu), 14 from Semen Lablab Album (SL, Biandou), 71 from Rhizoma Dioscoreae (RD, Shanyao), 133 from Fructus Jujube (FJ, Dazao), and 280 from Radix Glycyrrhizae (RG2, Gancao). A description of the plant species of eight QIHs is provided in Table 1. These herbs are commonly used to supplement Qi in Traditional Oriental Medicine. Table 2 compares the traditional use of eight QIHs in Korean, Chinese, and Vietnamese traditional medicine. In general, their uses in these three East Asian countries are similar; the eight QIHs are mainly used to treat organ Qi deficiency, fatigue, loss of appetite, long-term diarrhea, etc.

### 2.2. Screening of Unique Compounds

Of the 860 compounds, 19 common compounds were present in three or more herbs among the eight QIHs. Palmitic acid was the most common compound; it occurred in six of the eight QIHs. Stigmasterol, darutoside, nicotinic acid, and proline (prolinum) appeared in four QIHs. The remaining compounds appeared in three herbal groups, as shown in Table 3.

Within the group of common compounds, we found 13 unique compounds with a score >5.00, obtained using Equation (1) (Table 3). The identified unique compounds included six amino acids (proline, L-arginine [L-], L-alanine [LPG], glycine [GLY], aspartic acid [ASI], L-valine), two vitamins or vitamin-like molecules (nicotinic acid, choline), two lipids or lipid-like molecules (darutoside, mairin), and one carbohydrate (methose). Relevant unique compounds of QIHs in energy metabolism are shown in Figure 2.

### 2.3. Gene Ontology (GO) and Kyoto Encyclopedia of Genes and Genomes (KEGG) Pathway Enrichment Analyses

Using the TCMSP, BATMAN-TCM, and SwissTargetPrediction databases, 13 lists of targets of unique compounds were obtained, including 34 targets of darutoside, 152 targets of nicotinic acid, 114 targets of proline, 18 targets of 20-hexadecanoylingenol, 116 targets of 12-O-nicotinoylisolineolone, 148 targets of choline, 113 targets of mairin, 157 targets of L-arginine, 449 targets of L-alanine, 506 targets of glycine, 43 targets of methose, 447 targets of aspartic acid, and 427 targets of L-valine. 

After inputting the 13 lists of targets of unique compounds into the DAVID database, KEGG pathway enrichment analysis revealed 46 signaling pathways of darutoside, 25 of nicotinic acid, 11 of proline, 8 of 20-hexadecanoylingenol, 132 of 12-O-nicotinoylisolineolone, 27 of choline, 26 of mairin, 17 of L-arginine, 44 of L-alanine, 50 of glycine, 8 of methose, 44 of aspartic acid, and 44 of L-valine (*p* < 0.01). As shown in Figure 3, unique compounds were mainly involved in neuroactive ligand–receptor interaction, metabolic pathways, the calcium signaling pathway, arginine and proline metabolism, biosynthesis of amino acids, and arginine biosynthesis. The targets of the six amino acids and choline had many shared pathways. In addition, GO analysis showed that unique compounds were mainly involved in the responses to drugs, xenobiotic stimulus, ethanol, lipopolysaccharide, hypoxia, and glucocorticoid of the biological process category. Seven compounds were found to be involved in regulating the aging pathway. In the cell components category, the targets of the unique compounds were related to the components of mitochondria (mitochondrial inner membrane, mitochondrial outer membrane, and mitochondrial matrix) and neuronal cells (neuron projection, neuronal cell body, integral component of presynaptic membrane, postsynaptic membrane, synapse, dendrite, and perikaryon). Molecular functions showed identical protein binding, enzyme binding, oxidoreductase activity, and protein homodimerization activity (Figure 4).

To interpret the relationships between QIHs and their unique compounds and pathways, an herb-unique compound-target-pathway network was constructed. As shown in Figure 5, the network contained 8 QIHs, 13 unique compounds, 439 targets (proteins with no connection to the pathways were removed), and 13 main pathways.

## 3. Discussion

In TM, Qi plays an important role as a vital energy source in the growth of the human body and the physiological functioning of the organs and meridians [38]. QIHs are known to provide vital energy (Qi), fight fatigue, strengthen resistance, and promote health recovery. However, their unique action mechanisms have not yet been elucidated. Recently, a drug-centered approach employing bioinformatics provided an advanced scientific understanding of the TM theory [39]. In the present study, we proposed an integrated process that combines QIH identification, unique compound screening, and pathway enrichment analyses to understand the common mechanisms of QIHs that are unique from other traditional herbs. Our results revealed that unique compounds and their related pathways fundamentally play a critical role in the human body, and their functions are related to the concepts of Qi in TM.

In TM, herbs are grouped by their functional similarities, such as Qi-invigorating, blood- invigorating, or heat-clearing herbs, to treat specific syndromes of human diseases, including Qi deficiency, blood deficiency, and hot syndrome. In modern medicine, the effects of herbs or herbal groups are reflected by their chemical composition. Combining these ideas, we proposed a unique score index to screen for unique compounds to understand the underlying mechanism of the Qi-invigorating function. Among 860 compounds contained in QIHs, 13 unique compounds were identified with a unique score >5.00 (Table 3). This score represents the commonality of compounds in QIHs and the rarity of compounds in the other herbal groups: the higher the score, the more unique the compound. Therefore, we screened compounds with scores ranging from 2.00–10.00; finally, 5.00 was set as the cutoff point, indicating that the compound was at least five times more frequent in the QIH group than in other herbs.

The results show that most of the unique compounds are essential for the human body, including amino acids (proline, L-arginine, L-alanine, glycine, aspartic acid, L-valine) [40], vitamins or vitamin-like molecules (nicotinic acid, choline) [41,42], lipids or lipid-like molecules (darutoside, mairin) [43,44], and carbohydrates (methose) [45]. According to the theory of TM, Qi promotes energy conversion and substance metabolism, thereby promoting human growth and development, and also maintains the normal temperature of the human body [1]. Among the unique compounds, methose, darutoside, and mairin are carbohydrates and lipids, which are known as energy sources for the human body. Amino acids can also generate energy and are a vital source of nitrogen for a variety of macromolecules required by the body. Figure 2 shows six amino acids that participate in energy metabolism in different ways to produce intermediates such as pyruvic acid, α-ketoglutaric acid, succinyl coenzyme A, and oxaloacetate. In the tricarboxylic acid cycle, these intermediates are oxidized to carbon dioxide to generate energy [46]. Nicotinic acid is also involved in energy metabolism owing to the generation of NAD^+^, which is essential for energy metabolism and other metabolic processes [47]. Amino acid metabolism contributes to the regulation of body temperature by promoting muscle protein synthesis via insulin–mTOR-dependent activation [48,49] or by regulating central thermoregulatory control and peripheral heat production [50]. In summary, modern knowledge of the effects of unique compounds might be closely related to the energizing and warming effects of Qi in TM.

Besides their roles in the human body, unique compounds also play important roles in the metabolism of plants. Similar to humans, carbohydrates (methose), and lipids (darutoside, mairin) are used in plants as sources of energy essential to carrying out normal functions, such as growth and metabolism. Meanwhile, amino acids have a variety of crucial roles in plant metabolism. Aside from protein synthesis, amino acid metabolism is closely connected to the metabolism of energy, carbohydrates, hormones, and secondary metabolism, carbon/nitrogen budget, and stress responses [51]. Thus, they increase tolerance to hard conditions and diseases (Proline, Arginine, Aspartic acid), stimulate chlorophyll synthesis (Arginine, Alanine, Glycine), and influence growth velocity (Alanine, Valine) [52]. Further, nicotinic acid has been shown to enhance NADPH levels, which are important for plant antioxidant defense as well as growth [53]. Choline is an essential molecule in plants because it is required for the synthesis of the membrane phospholipid phosphatidylcholine; consequently, all plant and animal cells need choline to maintain structural integrity [54].

In addition, previous studies have shown that amino acids and vitamins of unique compounds help improve symptoms similar to Qi deficiency or traditional indications of QIHs (Table 2), such as increasing endurance, fighting fatigue [55,56,57,58,59,60], developing muscle mass [61,62,63,64], strengthening the immune system [65,66,67], and promoting normal appetite [68,69]. Amino acids can increase hormone production, modify fuel utilization, and prevent the side effects of overtraining and mental fatigue [55]. For instance, L-valine may inhibit the production of serotonin [55,56], L-arginine may alter blood lactate concentrations and metabolic indicators of respiration [57], and aspartic acid reduces ammonia accumulation [55] and serves as an excitatory neurotransmitter in the brain [58]. These mechanisms provide resistance to fatigue and lead to greater endurance. The combination of amino acids can improve lean muscle mass and strength [61], decrease muscle damage, and boost cytoprotective effects [62,63,64]. Glycine regulates immune function, the production of superoxide, and the synthesis of cytokines by altering intracellular Ca^2+^ levels [65], while L-alanine also strengthens the immune system by enhancing the production of antibodies [66,67]. Nicotinic acid has been shown to regulate appetite [68], and glycine controls the intake of food and behavior because it plays a crucial role as a neurotransmitter in the central nervous system [69]. Symptoms of Qi deficiency are similar to those of deficiency of the unique compounds. Amino acid deficiency causes fatigue, a depressed immune system, aging, weight loss, slow growth, diarrhea, and muscle weakness [70,71,72]. The effects of low choline consumption include muscle damage [73] and apoptotic death of lymphocytes [74], and a lack of nicotinic acid intake leads to nonspecific clinical symptoms, including weakness, appetite loss, fatigue, and apathy [41]. In summary, these unique compounds improve the symptoms of Qi deficiency, and the lack of these substances also leads to symptoms similar to those of Qi deficiency.

In this study, GO and KEGG pathway enrichment of unique compounds showed that most of the unique compounds predominantly participated in the neuroactive ligand–receptor interaction and the metabolism of amino acids in the KEGG pathway (Figure 3) and involved cellular components such as mitochondria, neurons, and their transmitters (Figure 4). In detail, six amino acids and choline participated in amino acid synthesis and metabolism, such as the metabolism of arginine, proline, tryptophan, tyrosine, glycine, serine, threonine, cysteine, and methionine, indicating the central role of amino acids and their pathways in the unique action mechanisms of QIHs. Numerous intermediates of energy transduction pathways are involved in amino acid production. Thus, cells can balance the degradation of compounds for energy mobilization and synthesis of starting materials for macromolecular construction [75]. The cellular component category of the GO analysis revealed unique compounds related to mitochondria, which play a crucial role in cellular energy metabolism for ATP production [76]. Previous research has found that QIHs enhance mitochondrial ATP production ability [6,77], levels of ATP, adenylate energy charge, and total adenylate pool [1]. In the present study, the KEGG and GO analyses also revealed unique compounds related to neuron composition that are involved in neuroactive ligand–receptor interactions. Some of the unique compounds show neuromodulatory effects, such as aspartic acid and proline, whereas others act as essential precursors to neurotransmitters, such as L-arginine, L-alanine, and L-valine [78]. Glycine and aspartic acid are known to have neurotransmitter-like effects; therefore, they can provide resistance to fatigue or control food intake [58,69,78]. Choline is required in the formation of essential membrane phospholipids and for the biosynthesis of the neurotransmitter acetylcholine [79]. In summary, the pathways of unique compounds are involved directly or indirectly in energy metabolism and neurotransmission, acting on various types of cells, such as neurons, and on specific cellular components, such as mitochondria.

Using network pharmacology, Jing et al. showed that the common pathway of Qi-invigorating prescription (including RG1, RAM, RG2, and *Poria cocos*) is involved in environmental information processing (neuroactive ligand–receptor interaction, calcium signaling pathway) and organismal systems (mainly the digestive system), rather than the metabolism pathway [80]. The similarities and differences with our study may have been due to the different input herbs. Moreover, our study focused on the unique compounds of the QIH group instead of analyzing the active ingredients of the Qi-invigorating prescription. In contrast, other studies focus on analyzing major compounds that are famous as valuable ingredients of different medicinal products. For example, the major bioactive components of *Panax ginseng* are the ginsenosides, a group of saponins with a dammarane triterpenoid structure [81], whereas astragalosides (I-VIII) are considered characteristic bioactive therapeutic compounds of *Astragali radix* [82]. These different approaches explore different perspectives on the effects of QIHs. 

This is the first study to investigate the unique compounds of a functional herbal group using a new bioinformatics-based approach. It provides a unique perspective of Qi in TM, identifies unique compounds of QIHs, and uncovers their underlying mechanism, thereby elucidating the underlying mechanism of Qi-invigorating effects. We hope that our study will help unify more aspects of Eastern and Western philosophy in medicine. This approach is expected to be applicable to other functional herbal groups and scientific elucidation of other concepts of TM. However, the present study has some limitations. Similar to other studies using a bioinformatics-based approach, the analysis results are highly dependent on the database’s input data. This study also lacks experimental verification, which will be addressed in future research.

## 4. Materials and Methods

### 4.1. Identification of QIHs and Their Compounds

To identify representative herbs in the Qi-invigorating group, QIHs were retrieved from three different textbooks written in different languages: “Zhong yao xue” (written in Chinese) [83], “Boncho-hak” (written in Korean) [30], and “Chinese medical herbology and pharmacology” (written in English) [84]. 

To identify the chemical compounds of common QIHs, their pinyin names were searched for in the Traditional Chinese Medicine Database and Systems Pharmacology Analysis Platform (TCMSP) database version 2.3 [85]. The molecular IDs and names of the compounds were noted for the next step.

### 4.2. Screening of Unique Compounds

To uncover the unique functions of the QIH group, unique compounds were screened. First, common compounds occurring in three or more herbs among the QIHs were chosen. Second, unique compounds showing frequent occurrence in QIHs but rare occurrence in all traditional herbs were chosen. To identify a unique compound, we proposed the following equation:(1)$$\mathrm{Unique}\;\mathrm{score}=\frac{\left[\mathrm{compound}\;\mathrm{occurrence}\;\mathrm{in}\;\mathrm{QIHs}\right]/\left[\;\mathrm{Total}\;\mathrm{number}\;\mathrm{of}\;\mathrm{QIHs}\right]}{\left[\mathrm{compound}\;\mathrm{occurrence}\;\mathrm{in}\;\mathrm{total}\;\mathrm{herbs}\right]/\left[\mathrm{Total}\;\mathrm{number}\;\mathrm{of}\;\mathrm{herbs}\right]}$$

In this equation, “compound occurrence in total herbs” is the number of occurrences of a compound in a total of 503 herbs in the TCMSP database. Compounds with a unique score > 5.00 were chosen as the unique compounds, and all the data were processed using Microsoft Excel 2019. Furthermore, their corresponding names, PubChem compound IDs, and Chemical Abstracts Service numbers were entered into the PubChem database (https://pubchem.ncbi.nlm.nih.gov/, accessed on 21 March 2022) to acquire the compound structures for target prediction.

### 4.3. GO and KEGG Pathway Enrichment Analyses Using Network Pharmacology Approach

To determine the action mechanisms of unique compounds, metabolic pathways were obtained using a network pharmacology approach. First, the targets of unique compounds were predicted using TCMSP version 2.3 [85], bioinformatics analysis tool for molecular mechanism of traditional Chinese medicine (BATMAN-TCM) [86], and SwissTargetPrediction databases [87]. The UniProt database was subsequently used to obtain the official gene symbol format of the “Homo sapiens” genes [88], and duplicate targets of the three databases were removed. Second, targets of each unique compound were imported into Database for Annotation, Visualization, and Integrated Discovery (DAVID) version 2021 [89] for GO and KEGG pathway enrichment analyses with *p*-value < 0.01 and species “Homo sapiens”. GO and KEGG pathway lists of unique compounds were compared to identify overlapping pathways that would represent the unique mechanism of QIHs. Finally, a network of “herb-unique compound-target-pathway” was constructed using Cytoscape 3.9.0 software to explore the relationship among herbs, unique compounds, their targets, and overlapping pathways [90].

## 5. Conclusions

This study proposes an informational approach to understand the unique mechanisms of Qi-invigorating effects by identifying the unique compounds of QIHs. Thirteen unique compounds of eight QIHs were identified, which mainly included amino acids and vitamins that are involved in energy metabolism and improve the symptoms of Qi deficiency. Qi-invigorating effects are mainly related to amino acid metabolism and neurotransmission, which occur in various components of the cell, such as mitochondria and neurons. These results help interpret the concept of Qi in a scientific manner and provide evidence regarding the treatment effectiveness of QIHs.

## Figures and Tables

**Figure 1 plants-11-02470-f001:**
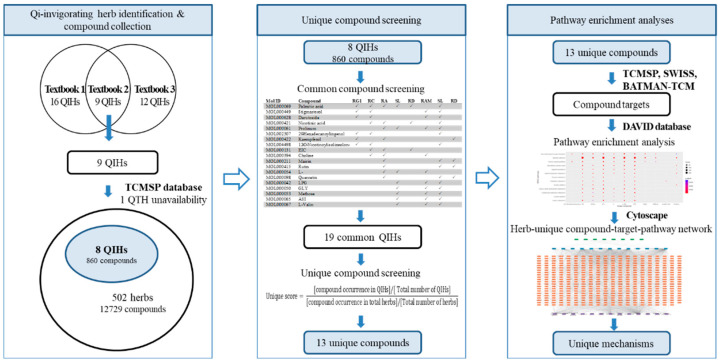
Flow chart of study.

**Figure 2 plants-11-02470-f002:**
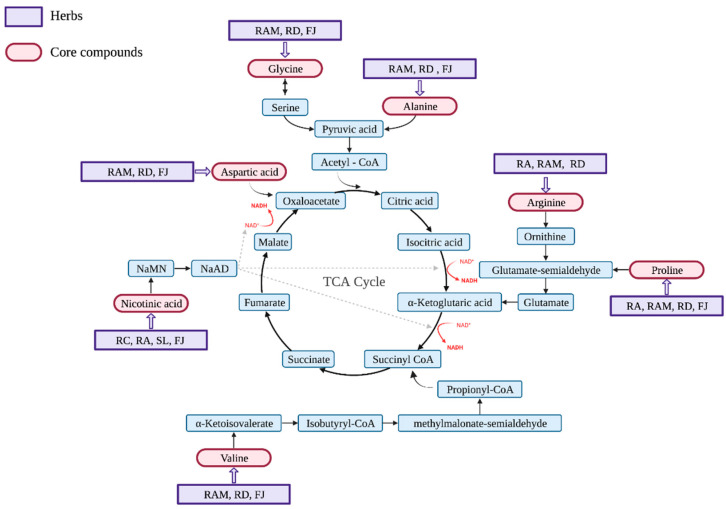
Summary of the energy metabolism of the unique compounds in humans. RA, *Radix Astragali*; RAM, *Rhizoma Atractylodis Macrocephalae*; RD, *Rhizoma Dioscoreae*; FJ, *Fructus Jujube*; RC, *Radix Codonopsis*; SL, *Semen Lablab Album*; TCA cycle, Tricarboxylic acid cycle; MaMN, nicotinic acid mononucleotide; NaAD, nicotinic acid adenine dinucleotide; CoA, coenzyme A.

**Figure 3 plants-11-02470-f003:**
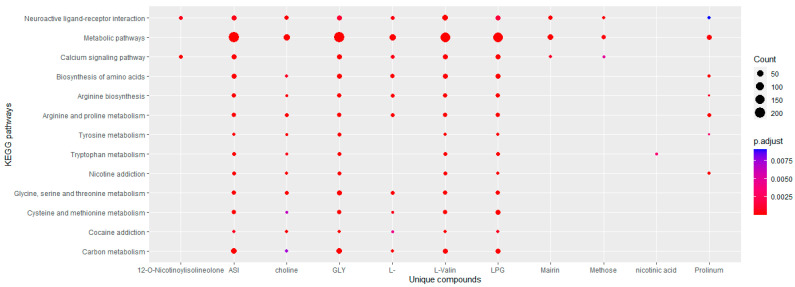
Top main KEGG pathways of unique compounds of Qi-invigorating herbs. LPG, L-alanine; GLY, glycine; ASI, aspartic acid; L-, L-arginine.

**Figure 4 plants-11-02470-f004:**
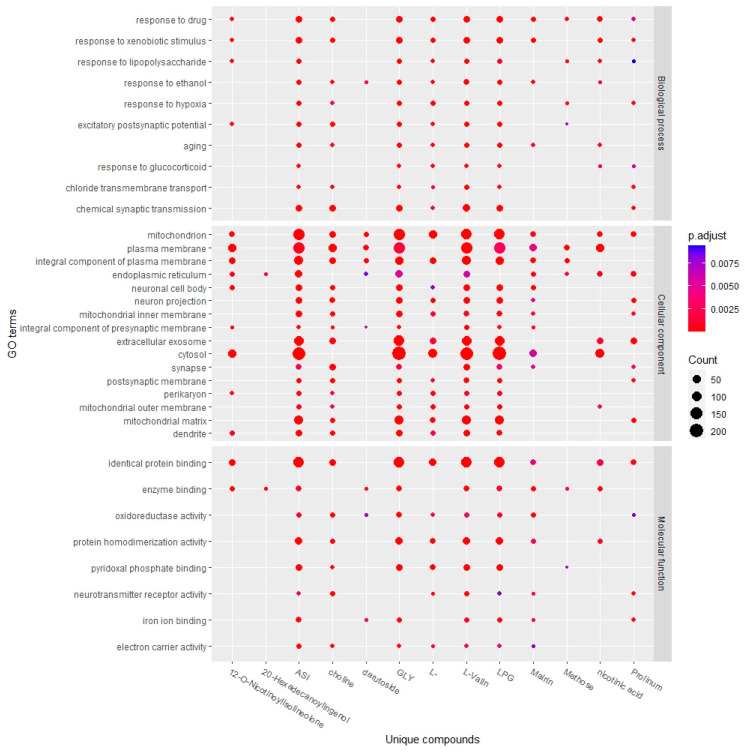
Top main GO terms of unique compounds of Qi-invigorating herbs. LPG, L-alanine; GLY, glycine; ASI, aspartic acid; L-, L-arginine.

**Figure 5 plants-11-02470-f005:**
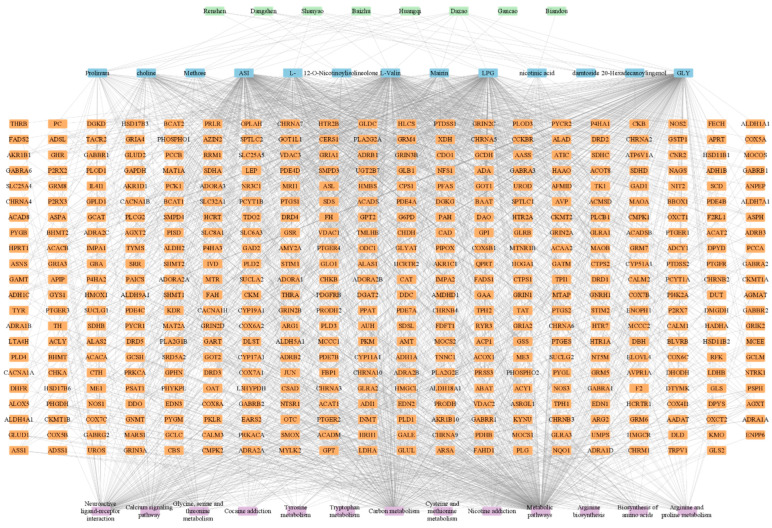
The herb-unique compound-target-pathway network. LPG, L-alanine; GLY, glycine; ASI, aspartic acid; L-, L-arginine.

**Table 1 plants-11-02470-t001:** Descriptions of the plant species of eight Qi-invigorating herbs.

Pharmactical Name	Plant Scientific Name & [Family]	Plant Description	Chemical Composition
Radix Ginseng	*Panax ginseng* C.A.Mey. [Aralilaceae]	A deciduous perennial plant, growing in the mountains of the Korean Peninsula, Northeast China, and Russian Far East [13].	Ginsenosides are major active compounds with anti-aging, anti-inflammatory, antioxidant, and anticancer effects [14].
Radix Codonopsis	*Codonopsis pilosula* (Franch.) Nannf. [Campanulaceae]	A perennial species of flowering plant, widely cultivated in Northwest China, and primarily distributed from 1500 to 3000 m altitude [15].	Polysaccharides, flavonoids, alkaloids, triterpenoids, lignans, glycosides, lactones, and sesquiterpenes [15].
Radix Astragali	*Astragalus membranaceus* (Fisch.) Bge. var. *mongholicus* (Bge.) Hsiao. [Leguminosae]	A flowering plant, often growing on hillsides, beside ditches, or in woodlands in Heilongjiang, Jilin, Liaoning (China), Tibet, and Inner Mongolia [16,17].	Polysaccharides, saponins, flavonoids, amino acids, and other compounds [17].
Rhizoma Atractylodis macrocephalae	*Atractylodes macrocephala* Koidz. [Asteraceae]	A perennial herb, mainly distributed in mountainous wetlands, such as Sichuan, Yunnan, and Guizhou in China [18].	Lactones, polysaccharides, volatile oil, amino acids, vitamins, and resins [19].
Semen Lablab Album	*Lablab purpureus* (L.) Sweet [Fabaceae]	A bean species and perennial vine, widely distributed in tropical and subtropical countries [20].	Steroids, essential oils, alkaloids, tannins, flavonoids, saponins, coumarins, terpenoids, pigments, glycosides, anthnanoids, and minerals [20].
Rhizoma Dioscoreae	*Dioscorea opposita* Thunb. [Dioscoreaceae]	A perennial climbing vine, native to China and cultivated in Asia [21].	Polysaccharides, amino acids, mannan, allantoin, dopamine, glucoprotein, choline, ergosterol, campesterol, and saponins [22].
Fructus Jujube	*Ziziphus jujuba* Mill. [Rhamnaceae]	A small deciduous tree or shrub, growing mostly in Europe, Southeast Asia, and Australia [23].	Vitamin C, phenolics, flavonoids, triterpenic acids, and polysaccharides [23].
Radix Glycyrrhizae	*Glycyrrhiza uralensis* Fisch. [Fabaceae]	A flowering plant, widely distributed throughout Inner Mongolia, Gansu, Xinjiang, Qinghai, Shaanxi, Ningxia, Shanxi, and Heilongjiang [24].	Flavonoids, triterpenoid saponins, coumarins, phenols and polysaccharides [24].

**Table 2 plants-11-02470-t002:** Comparison of traditional use of 8 Qi-invigorating herbs in Korean, Chinese and Vietnamese medicine.

Pharmactical Name	Traditional Medicinal Uses in Korea	Traditional Medicinal Uses in China	Traditional Medicinal Uses in Vietnam
Radix Ginseng	General weakness, acute vomiting with diarrhea, hiccups and vomiting, and phlegm [25].	Collapse caused by Qi deficiency, fatigue, low appetite, diarrhea, shortness of breath, spontaneous perspiration, diabetes, amnesia, insomnia, and impotence [26].	Collapse caused by Qi deficiency, cold limbs, low appetite, diabetes, frailness caused by long-term illness, and fright palpitation [27].
Radix Codonopsis	Dyspepsia, fatigue and respiratory disease [28]. *C. pilosula* is also proverbially consumed as a putative functional food [15].	Qi deficiency of the spleen and lungs, shortness of breath, cough, anorexia, loose stools, palpitation, and wasting-thirst. Used as a substitute for *Radix ginseng* [15].	Qi deficiency of the spleen and the lungs, low appetite, asthenic dyspnea and cough, heart palpitations and shortness of breath, and internal heat diabetes [28]. Used as a substitute for *Radix ginseng* [29].
Radix Astragali	Spontaneous or night sweating, prolapse of rectum, diarrhea, and edema [30].	To treat general weakness and chronic illness and increase overall vitality [17].	Lack of strength, loss of appetite, sunken middle qi, chronic diarrhea, prolapse of rectum, spontaneous sweating, wasting-thirst [27].
Rhizoma Atractylodis Macrocephalae	Used as a diuretic and stomachic drug [31].	Abdominal distension, loss of appetite, dizziness, upset, fetal movement, and spontaneous sweating [18].	Abdominal distension and diarrhea, edema, spontaneous sweating, and threatened miscarriage [27].
Semen Lablab Album	Spleen-stomach weakness, chronic lack of appetite, vomiting and diarrhea, and edema [30].	Spleen-stomach weakness, no appetite, sloppy stool, profuse white vaginal discharge, vomiting and diarrhea caused by summer heat-dampness, oppression in the chest, and abdominal distension [32].	Spleen-stomach weakness, no appetite, sloppy stool, profuse white vaginal discharge, vomiting, and diarrhea [27].
Rhizoma Dioscoreae	Spleen deficiency and decreased appetite, long-term diarrhea, lung deficiency asthma and cough, frequent urination, and wasting-thirst [30].	Poor appetite, chronic diarrhea, asthma, dry coughs, frequent or uncontrollable urination, and diabetes [33].	Reduced food intake, chronic diarrhea, dyspnea and cough by lung deficiency, seminal emission, abnormal vaginal discharge, and wasting-thirst [27].
Fructus Jujube	Lack of appetite, nervous asthenia and hysteria, biliousness, and bronchitis [34].	Spleen deficiency, decreased appetite, loose stool, fatigue and lack of strength, blood deficiency and sallow complexion, women’s hysteria, and restlessness of mind and will [35].	Reduced food intake caused by spleen deficiency, lack of strength, sloppy stool, and hysteria [27].
Radix Glycyrrhizae	Wasting-thirst [36].	To invigorate the qi of the heart and spleen, and harmonize the characteristics of other herbs [37].	Spleen-stomach weakness, fatigue, palpitations, and swelling abscess [27].

**Table 3 plants-11-02470-t003:** Common and unique compounds of eight Qi-invigorating herbs.

Mol ID	Compound	RG1	RC	RA	RAM	SL	RD	FJ	RG2	Occurrence in QIHs	Occurrence in TCMSP	US	UC
MOL000069	Palmitic acid	✓	✓	✓	✓	✓		✓		6	240	1.57	
MOL000449	Stigmasterol	✓	✓				✓	✓		4	133	1.89	
MOL000628	Darutoside	✓	✓				✓	✓		4	16	15.69	✓
MOL000421	Nicotinic acid		✓	✓		✓		✓		4	23	10.91	✓
MOL000061	Prolinum			✓	✓		✓	✓		4	14	17.93	✓
MOL002307	20-Hexadecnoylingenol	✓	✓					✓		3	11	17.11	✓
MOL000422	Kaempferol	✓		✓					✓	3	133	1.42	
MOL004498	12-O-Nicotinolisolineolone	✓	✓					✓		3	5	37.65	✓
MOL000131	EIC		✓	✓		✓				3	154	1.22	
MOL000394	Choline		✓	✓			✓			3	24	7.84	✓
MOL000211	Mairin			✓				✓	✓	3	29	6.49	✓
MOL000415	Rutin			✓				✓	✓	3	76	2.48	
MOL000054	L-			✓	✓		✓			3	12	15.69	✓
MOL000098	Quercetin			✓				✓	✓	3	188	1.00	
MOL000042	LPG				✓		✓	✓		3	7	26.89	✓
MOL000050	GLY				✓		✓	✓		3	13	14.48	✓
MOL000053	Methose				✓		✓	✓		3	9	20.92	✓
MOL000065	ASI				✓		✓	✓		3	13	14.48	✓
MOL000067	L-Valine				✓		✓	✓		3	12	15.69	✓

RG1, Radix Ginseng; RC, Radix Codonopsis; RA, Radix Astragali; RAM, Rhizoma Atractylodis Macrocephalae; SL, Semen Lablab Album; RD, Rhizoma Dioscoreae; FJ, Fructus Jujube; RG2, Radix Glycyrrhizae (RG2, Gancao); QIHs, Qi-invigorating herbs; US, Unique score; UC, Unique compound; LPG, L-alanine; GLY, glycine; ASI, aspartic acid; EIC, linoleic acid; L-, L-arginine.

## Data Availability

The data used to support the findings of this study are available from the corresponding author upon request.

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
