# Peer review of "Molecular Mechanisms Underlying Qi-Invigorating Effects in Traditional Medicine: Network Pharmacology-Based Study on the Unique Functions of Qi-Invigorating Herb Group"

_plants, 2022, doi:10.3390/plants11192470_

Round 1

Reviewer 1 Report

My comment: 1 This is a significant proposal. The authors designed three steps to provide evidence for Qi tonifying herbs (QTHs) which search the thirteen unique compounds on QTHs from multiple popular databases, screen the special features from the unique compounds and identify the chemical performance function in the microcirculation of these unique compounds.

2 Your research can confirm the clinical effect of Traditional Qi-tonifying herbs on participating energy metabolism and improving qi deficiency syndromes, so you provide the evidence regarding the QTHs treating effect in the clinic. 

3 Whether will you go further research: as a traditional herbal formula with the way which is commonly used, like the Four Gentlemen formula:  Renshen, Baizhu, Fuling and Gancao, whether can produce a multipliable effect if these herbs are used together, compared with used by any QTHs.

4 Qi-tonifying: I wonder whether Tonify is a proper English word? Tonify as a word is few used in the UK and can't be found in the dictionaries of the West, it is also marked as a wrong word in this editing system. Whether will you consider: Qi-invigorating, or Qi-reinforcing? 

Many thanks.

Author Response

We are re-submitting our paper "Molecular Mechanisms Underlying Qi-invigorating Effects in Traditional Medicine: Network Pharmacology-based Study on the Unique Functions of Qi-invigorating Herb group." We appreciate the time and effort that the editor and the reviewers have dedicated to providing valuable feedback on our manuscript. Overall, we agree with the comments of the reviewers. We believe that our paper's revised version addresses all concerns the reviewers raised in detail. We complied with the Plants manuscript requirement, and every change to the manuscript has been documented. Please find the detail in the attached files. The reviewer's comments are in black, and the authors' responses are in red.

Reviewer 2 Report

Undoubtedly, it is important to find answers - why the sources of Qi are the life energy. What plants and why contribute to its restoration and maintenance. However, this question (problem) is very extensive, and requires the study of a large number of small issues and details of chemical processes both in plants and in the human body.

In the water part of the article, it is important to indicate the role of various metabolites synthesized in plants and indicate their role in human life. These are, first of all, vitamins, organic acids, flavonoids, phenolic compounds, and so on.

All plants examined must be named and given their full Latin names. It is necessary to give a brief description of each of the plant species that were used in the work, including their chemical composition.

It is important to indicate not only that these species are mentioned in the reference books on medicinal plants in Korea, China, Vietnam, but also to give brief specific information for which diseases they are used in each country. This information should be presented in a tabular form, making an analysis of the similarity of the data or differences in the use of these types.

The methodological part of the work must contain information about the source material, where and when it was collected, by whom it was determined, and where the herbarium specimens of these species are stored.

Author Response

(The authors gave the same response as above.)
